# Well-Ordered 3D Printed Cu/Pd-Decorated Catalysts for the Methanol Electrooxidation in Alkaline Solutions

**Karolina Kołczyk-Siedlecka \***, **Dawid Kutyła**, **Katarzyna Skibińska, Anna Jędraczka, Justyna Palczewska-Grela and Piotr Żabiński**

Faculty of Non-Ferrous Metals, AGH University of Science and Technology, Mickiewicza 30,
30-059 Krakow, Poland; kutyla@agh.edu.pl (D.K.); kskib@agh.edu.pl (K.S.); kwiec@agh.edu.pl (A.J.);
palczews@agh.edu.pl (J.P.-G.); zabinski@agh.edu.pl (P.Ż.)
\* Correspondence: kkolczyk@agh.edu.pl

**Abstract:** In this article, a method for the synthesis of catalysts for methanol electrooxidation based on additive manufacturing and electroless metal deposition is presented. The research work was divided into two parts. Firstly, coatings were obtained on a flat substrate made of light-hardening resin dedicated to 3D printing. Copper was deposited by catalytic metallization. Then, the deposited Cu coatings were modified by palladium through a galvanic displacement process. The catalytic properties of the obtained coatings were analyzed in a solution of 0.1 M NaOH and 1 M methanol. The influence of the deposition time of copper and palladium on the catalytic properties of the coatings was investigated. Based on these results, the optimal parameters for the deposition were determined. In the second part of the research work, 3D prints with a large specific surface were metallized. The elements were covered with a copper layer and modified by palladium, then chronoamperometric curves were determined. The application of the proposed method could allow for the production of elements with good catalytic properties, complex geometry with a large specific surface area, small volume and low weight.

**Keywords:** electroless metallization; catalysts; 3D printing

## 1. Introduction

Additive manufacturing is a technique that is increasingly used in new technologies [1–3]. The development of the digital revolution and 3D printing methods are changing the methods used for the production of functional objects. The use of 3D printing techniques allows for a quick transition from a digital model to a physical object. This ensures great flexibility in adapting a given geometry, as opposed to classical production methods like machining or casting for metallic parts or injection molding for plastics [4]. The advantage of this technique is the possibility of producing elements from materials such as plastic, metal or materials based on composites [5–7]. In addition, the modification of elements produced by the 3D printing method is also carried out, for example by covering them with metallic coatings. Thanks to the listed advantages and many possibilities, additive manufacturing has found many applications in many fields, for example in new technologies, medicine [8], catalysis [9–11] and electrochemistry [12–17].

Direct methanol fuel cells (DMFCs) are a promising solution for the problems of energy conversion. They are characterized by their small size, high energy conversion efficiency, low working temperature and the availability of methanol as fuel [18]. However, there are some limitations to introducing these cells into commercial markets. This is related to the low efficiency of the anode reaction due to the slow kinetics of methanol electrooxidation and the destruction of the electrode surfaces [19–21].

Good catalytic properties characterize noble metals such as Pt [22–24], Pd [25–27] and Ag [28–30]. However, these metals are expensive, and this limits their application possibilities.

Copper is not a very popular catalyst for the electrooxidation process due to its high sensitivity and its reactivity towards oxygen. Nevertheless, there are research works focused on the application of this metal or its alloys. The catalytic properties of other metals, such as nickel [31], copper [32,33] and their alloys Ni-Cu [34,35] have been analyzed. Catalysts synthesized with noble metals, e.g., Pd-Ni [36–39], Pt-Ni [22,40], Pd-Cu [26,41–43], Pt-Cu [44–47], have also been tested.

In the literature, catalytic coatings have often been deposited on conductive substrates such metals of graphite. Unfortunately, these materials often have a high density or they are more challenging to manufacture, especially with high-precision techniques. One of the solutions is the combination of additive manufacturing in terms of the possibility of obtaining complex geometry inside an element and the process of electroless deposition of metallic coatings [48]. In this paper, the results of research related to the synthesis of Cu/Pd coatings by electroless deposition on a resin substrate dedicated to 3D printing are presented. A novelty in this work is the development of electroless deposition on a complex surface. In our previous work [49,50], metallic electroless coatings were obtained on a flat surface. The obtained materials were analyzed in terms of catalytic properties in the methanol electrooxidation reaction. The presented experimental works are intended to develop a new type of catalysts.

## 2. Materials and Methods

In the experimental work, chemicals with analytical purity were used (ChemLand company, Poland). The main stage of research was the metallization of light-hardened resins. The substrate was made of FormLabs Form 2 (CadXpert company, Poland) resins (Clear and Gray) dedicated to the stereolithography (SLA) method. Both of them are characterized by chemical resistance, as described by the manufacturer [51,52].

A liquid resin with a volume of 1 mL was dropped onto a glass substrate, and then cured using a UV lamp (power 48 W) for 1 min. The cured samples were washed in two stages in isopropanol for 20 min every step, according to the manufacturer's instructions. Then, they were washed with demineralized water and dried.

The cured samples were degreased in 5 wt% NaOH solution at 70 °C for 10 min to remove other impurities from the surface. Then, the samples were washed in demineralized water and etched for 1 min at 70 °C in a chromic acid solution prepared as follows: 50 g $Cr_2O_3$, 1500 g $H_2SO_4$, 250 g $H_2O$. The chromium(IV) ions were neutralized and removed from the surface in 5 wt% HCl and 5 wt% $K_2S_2O_5$ solution for 3 min at room temperature. Then, the samples were washed in demineralized water. The surface of the resin was activated by Pd(II) ions, the samples were placed into 1 g/L $PdCl_2$ solution for 30 min, and during this time the ions were adsorbed on resin surface. After this step, the samples were placed into freshly prepared 20 g/L $NaBH_4$ solution to reduce the palladium ions.

After activation and reduction, the samples were thoroughly washed with demineralized water and placed into the metallization solution: 10 g/L $CuSO_4 \cdot 5H_2O$, 10 g/L tartaric acid, 10 mL/L formalin. The pH of the solution was modified by NaOH, and it was equal to 12. The temperature of solution was 40 °C. The copper was deposited for 10 and 20 min. Then, the samples were washed using demineralized water and placed for galvanic displacement into 2 g/L $PdCl_2$ solution for 30 s to 5 min. To compare the parameters of electroless deposited coatings with bulk material, the copper sheets were placed in the same solution for 1 and 5 min.

All samples were tested electrochemically in 0.1 M NaOH + 1 M methanol solution in a three-electrode system at room temperature. The first electrode was a Cu/Pd sample, the counter electrode was a Pt sheet, and as the reference electrode the saturated calomel electrode (SCE) was used. Cyclic voltammetry experiments were performed in the potential range between hydrogen and oxygen evolution with different scan rates.

Using SolidWorks 2018 SP 4.0 software and the FormLabs Form 2 3D printer, cylinder-shaped elements with a diameter of 2 cm and a height of 3 cm were designed and printed. The elements were characterized by large specific surface in relation to the dimensions.

Similar to the previous samples, the elements were metallized by copper and decorated by palladium. Cyclic voltammogram measurements and a chronoamperometric curve were performed to evaluate catalytic properties. As was carried out previously, the elements were tested in 0.1 M NaOH + 1 M CH3OH solutions at room temperature in the three-electrode system. The working electrode was the metallized element and the counter electrode was a Pt sheet. The cycling voltammograms were detected in the range from −1.0 to 1.5 V vs. SCE.

The obtained coatings were analyzed using scanning electron microscopy (SEM) with a JEOL—6000 Plus. The chemical composition and distribution of elements were determined using energy dispersive X-ray spectroscopy (EDS) analysis. The catalytical tests were performed using a BioLogic SP-200 potentiostat.

## 3. Results and Discussion

### 3.1. Physical Characterization

In this work, the concentration of copper and palladium in the obtained coatings was determined. For the selected sample, the analysis of Cu and Pd distribution was performed. In the main part of the research work, a series of catalytic tests for the methanol oxidation reaction were performed. SEM pictures of the coatings before and after catalytic tests were taken.

The elemental composition of the coatings obtained as a result of electroless (EL) copper deposition and galvanic displacement by palladium is shown in Figure 1. The copper concentration is marked in blue and the palladium concentration in red. The different types of obtained copper samples are marked with different symbol shapes. The change in Pd concentration was linear with the time of galvanic displacement. The palladium concentration was not dependent on the Cu deposition time. In both cases of 10 min and 20 min of electroless Cu metallization, the Pd content ranged from about 2.75% (30 s) to 16.5% by mass (5 min). In the case of Pd galvanic displacement on metallic copper, the palladium content after 5 min was equal to 15.12% by mass and on the Cu coatings this value was approx. 16.8% and 16.6% on the copper deposited for 10 and 20 min, respectively.

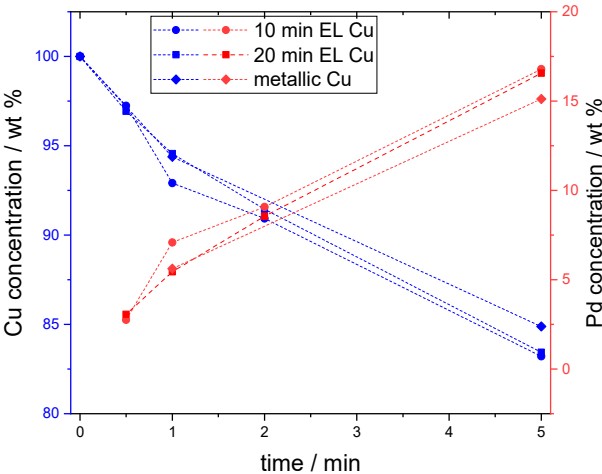

**Figure 1.** Composition of coatings depending on galvanic displacement time.

The SEM image taken for the coating obtained after 10 min of electroless Cu metallization and 5 min of galvanic exchange of palladium is presented in Figure 2. It can be seen that a smooth surface was obtained, and a slight precipitation can be observed. The mapping analysis showed that the distribution of Pd was homogeneous.

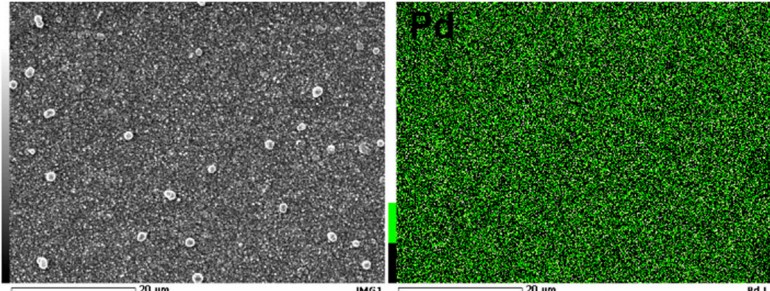

**Figure 2.** Mapping analysis of Cu/Pd after 10 min of electroless deposition of Cu and 5 min of galvanic displacement Pd. Magnification: ×2000.

### 3.2. Electrooxidation of Methanol on the Cu/Pd-Decorated Coatings

The methanol electrooxidation activity of Cu electrolessly deposited coatings modified by palladium was investigated in 0.1 M NaOH + 1 M CH$_3$OH solutions at room temperature using electrochemical techniques. Figure 3 presents cyclic voltammograms of materials, depending on electroless deposition of Cu and Pd galvanic displacement time (Figure 3a,b). As shown the presented cyclic voltammograms, a change in the anodic peak connected with the oxidation of methanol was observed. Depending on the time of Pd deposition by galvanic displacement and thus the different concentration, the maximum potential value changed, ranging from 0.8 V vs. SCE for 5 min Pd (about 16 wt. %) to 1.2 V vs. SCE after 2 min of Pd (about 10 wt. %) on the Cu substrate deposited for 10 min (Figure 3a). In the case of the Cu coatings deposited for 20 min, the anode peaks varied to a lesser extent, from 1.2 to 1.36 V vs. SCE (Figure 3b), with the highest activity observed for the modification by Pd for 30 s. To compare the catalytic properties, the diagrams for materials obtained on metallic copper are also presented (Figure 3c). Experiments were carried out on a copper substrate modified by palladium for 1 min and 5 min. No significant difference was observed between the measurements carried out; the potential of the anode peak is 1.3 V vs. SCE (Figure 3c).

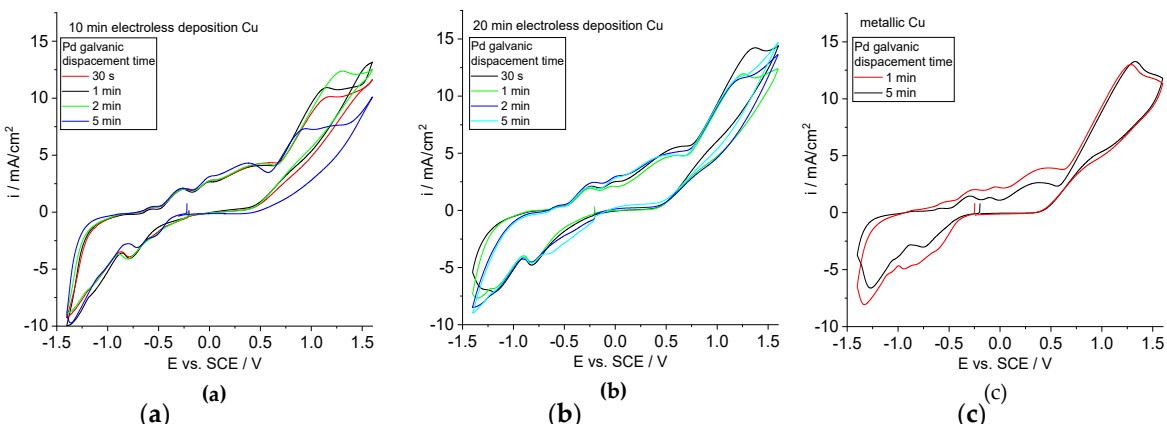

**Figure 3.** Cyclic voltammograms of Cu/Pd-decorated coatings depending on the substrate type and Pd galvanic displacement time. The substrates were as follows: 10 min (**a**), 20 min (**b**) electrolessly deposition copper and metallic copper (**c**). The measurements were performed in 0.1 M NaOH + 1 M methanol solution, scan rate: 50 mV/s.

The cyclic voltammogram curves for different scan rates were determined. Figure 4 shows them for two exemplary samples—the coatings obtained in 20 min Cu electroless metallization + 30 s and 5 min Pd galvanic displacement. As the scan rate increases, the anode peak from the oxidation of methanol increases. The same tendency was shown in all the analyses. Cyclic voltammograms of the remaining samples depending on the scan rates are presented in the Supplementary Materials (Figures S1–S6).

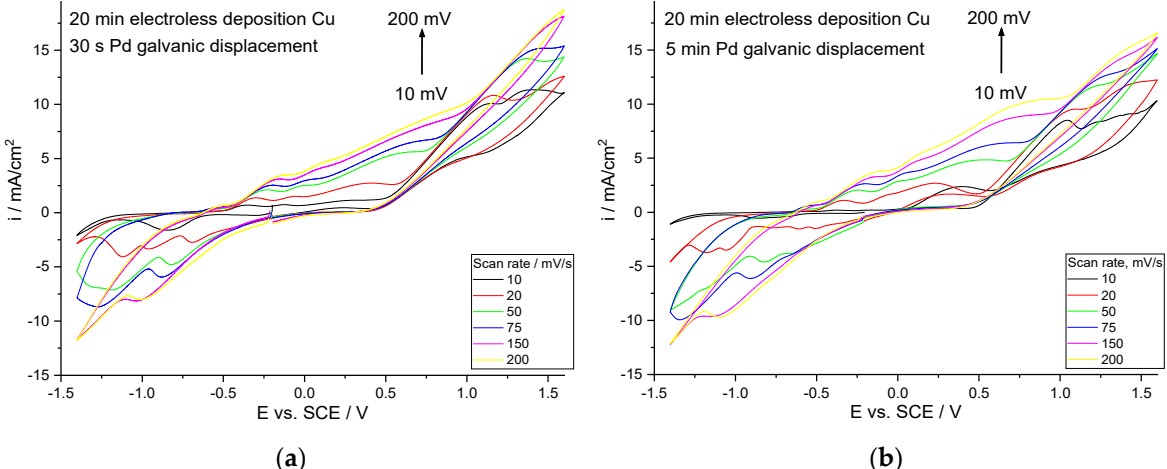

**Figure 4.** Cyclic voltammograms of obtained coatings in 0.1 M NaOH + 1 M methanol solution at different scan rates. The deposition parameters: 20 min electroless deposition of Cu, modification by palladium for 30 s (**a**) and 5 min (**b**).

Based on the cyclic voltammogram curves presented in Figure 4 and in the Supplementary Materials (Figures S5 and S6), the change in the relationship between the current density and the square root of the scan rate was determined (Figure 5a). This relationship has been shown to be linear, therefore it can be concluded that the methanol electrooxidation reaction takes place through diffusion control [44]. For all cases, a high correlation coefficient $R^2$ was obtained. Figure 5b shows that the potential changed linearly with the logarithm of the scan rate, suggesting that methanol oxidation on the surfaces is an irreversible process [44].

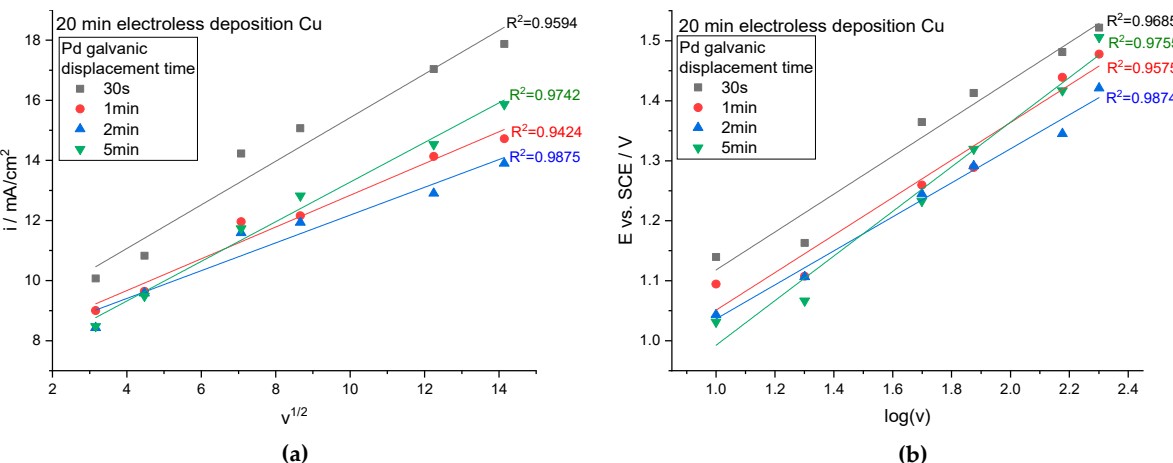

**Figure 5.** Characteristics of the coatings depending on the Pd galvanic displacement time. Change in anodic peak current with the square roots of scan rate (**a**), change in anodic peak potential with logarithm of scan rate (**b**). The copper was deposited for 20 min and modified by Pd.

The SEM pictures of the coatings prepared for electrochemical analysis and after catalytic tests are presented in Figure 6. The Cu coatings obtained electrolessly were characterized by continuity. In the case of copper deposited for 10 min, small holes were observed. Small irregularly shaped or spherical precipitates were observed on the coating surfaces. Precipitations came from the electroless copper deposition process, and not from Pd galvanic displacement, which can be observed in the mapping analysis (Figure 2) and in comparison to the Cu coatings obtained after 20 min (Figure 6). In the case of Cu deposited for 20 min, the coatings had a homogeneous morphology and small crystalline coatings

were observed. After catalytic tests, in the case of Cu/Pd materials obtained after 10 min of electroless copper deposition, the destruction of the coatings was clearly visible.

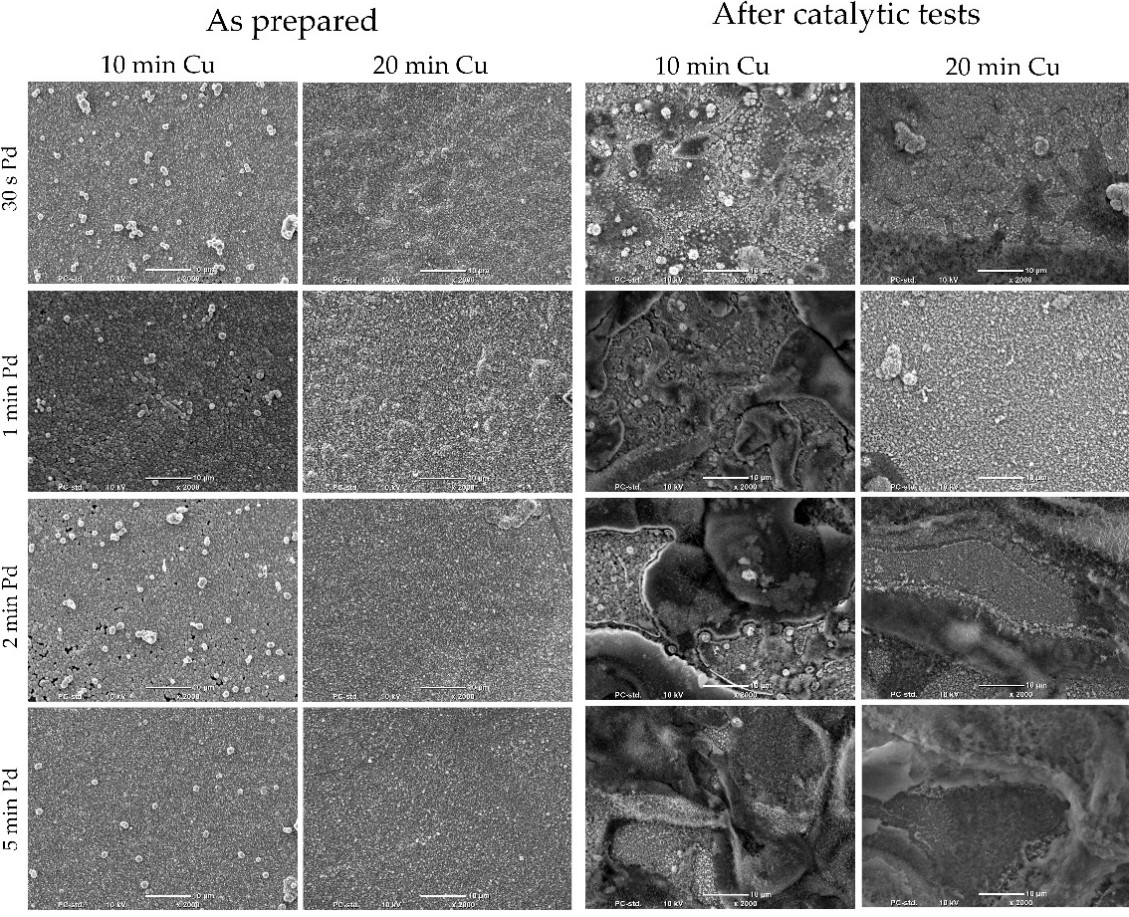

**Figure 6.** SEM pictures of electroless-deposited Cu modified by Pd, depending on copper deposition and galvanic displacement time as prepared and after electrochemical tests, magnification ×2000.

In some cases (e.g., after 30 s of Pd galvanic displacement) characteristic precipitations were still visible. Coatings based on Cu deposited for 20 min were characterized by less damage of the coatings. After the tests, the coatings were still stable and showed good adhesion. The bluish discoloration of the precipitate was clearly visible, indicating copper oxidation.

To assess the quality of coatings prepared on metallic copper, the SEM pictures were taken (Figure 7). Slight cracks were visible on the surface, and slight irregular precipitations were visible after 5 min of palladium deposition. Surface destruction was observed after catalytic tests—in the case of the coating deposited after 1 min with Pd galvanic displacement, the size of cracks on the surface increased. In the case of a higher palladium content, most of the precipitates from the surface disappeared. As with the resin-deposited coatings, it was observed that the coatings were oxidized after the tests.

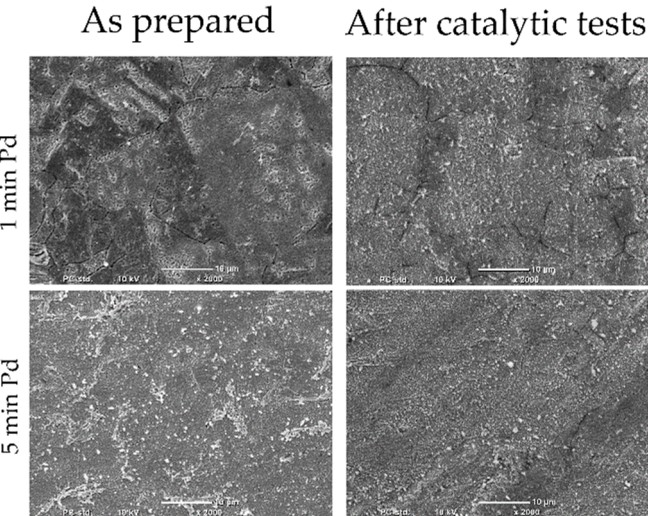

**Figure 7.** SEM pictures of metallic Cu modified by Pd, depending on galvanic displacement time as prepared and after electrochemical tests, magnification ×2000.

*3.3. Methanol Electrooxidation on 3D Printed Elements*

A prototype of a catalyst for the electrooxidation of methanol in basic solutions was made, and its design is presented in Figure 8. The cylindrical prints were designed based on common geometry [11]. Round holes were made parallel to and perpendicular to the cylinder's axis.

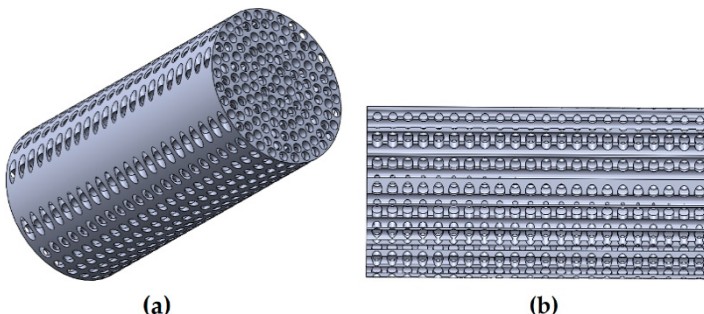

(a)          (b)

**Figure 8.** Drawing (**a**) and cross-section (**b**) of the catalyst made in the SolidWorks software.

Figure 8 shows the digital design of the element. The cross-section shown in Figure 8b shows that the element is characterized by high porosity, thanks to which it was possible to obtain a large specific surface, which is equal to 230 cm$^2$.

Two elements were prepared and metallized by copper according to the procedure described previously; the Cu metallization time was equal to 20 min. Then, the metallized elements were placed into 2 g/L PdCl$_2$ solutions for 2 and 5 min. The concentration of palladium in Cu/Pd coatings was determined by EDS.

The anodic peak was determined from the cyclic voltammogram curves presented in Figure 9a. The obtained potential was chosen for potentiostatic measurements, which were performed over 15 min in the same solutions. In case of the coatings modified by Pd for 5 min, the maximum determined potential was equal to 1.3 V vs. SCE. An anode potential of 1.05 V vs. SCE was observed on the curve of the coating deposited for 20 min with Cu and modified for 2 min by Pd. For the chronoamperometry curves, the stability of the current over time was observed (Figure 9b). After a certain period, both curves tended to plateau, when the Pd concentration was near 20% and over 10% by mass.

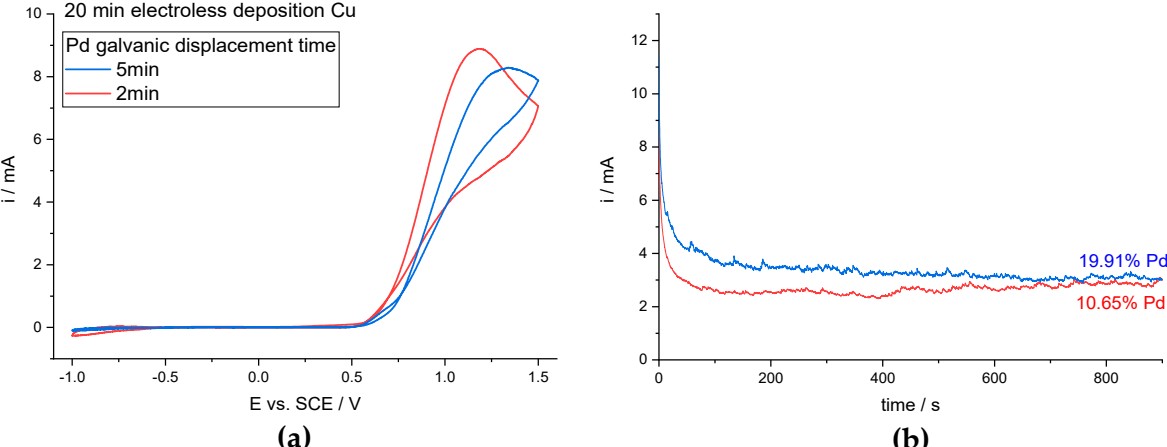

**Figure 9.** Cyclic voltammograms (**a**) and chronoamperometry (**b**) of metallized 3D prints. The Cu substrate was deposited in 20 min and modified by Pd in galvanic displacement in 2 and 5 min.

Figure 10 presents SEM pictures of Cu/Pd coatings obtained on 3D prints, depending on the modification time with palladium, as prepared and after catalytic tests. The coatings were characterized by a similar morphology, regardless of the palladium galvanic displacement time. Fewer minor precipitates were also observed for the shorter time of Pd deposition. The coatings did not deteriorate after catalytic tests; they remained compact and adhered to the substrate. Fewer precipitates on the surface were observed.

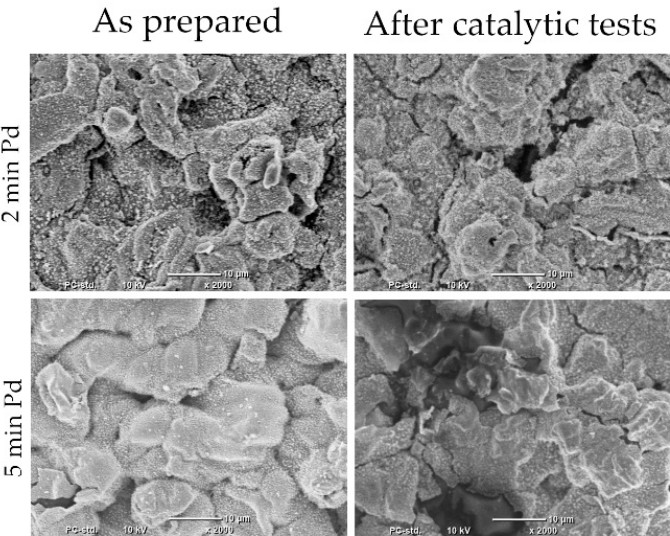

**Figure 10.** SEM pictures of coatings deposited on 3D prints as prepared and after catalytic tests, magnification ×2000.

## 4. Conclusions

The aim of this work was to develop a prototype of a catalyst intended for direct methanol fuel cells. A catalyst synthesis based on 3D printing and electroless copper deposition was proposed.

Metallic Cu coatings were modified by palladium through the galvanic displacement process. In the first stage, Cu/Pd coatings on a flat substrate were synthesized. The obtained coatings were characterized by a homogeneous distribution of elements on their surface. Cyclic voltammogram curves were obtained to determine the catalytic properties of Cu/Pd coatings. The coatings formed on the Cu substrate deposited for 20 min were characterized by smaller anode peak potential differences, depending on composition. By determining the linear relationship of currents in the anode peak as a function of the

element from the scan rate and the potential from log (v), it was determined that the reaction was diffusion-controlled and irreversible. Based on the SEM pictures, it was determined that coatings obtained on electroless copper deposition for 10 min are porous and unstable—after catalytic tests, the destruction of the coatings was visible. Coatings based on 20 min EL Cu were less damaged.

On the basis of the results obtained, the metallization parameters of substrates constituting 3D prints with a large specific surface were determined. Composition, cyclic voltammograms and chronoamperometry curves were determined and SEM pictures of the coatings were taken before and after the electrochemical tests. It was found that the coatings were not significantly damaged by the applied potential. Using these techniques, a 3D printed catalyst model was pre-developed.

**Supplementary Materials:** The following are available online at https://www.mdpi.com/2227-708 0/9/1/6/s1, Figure S1: Cyclic voltammograms of 10 min electroless (EL) copper + 30 s Pd in 0.1 M NaOH + 1 M methanol solution at different scan rates; Figure S2: Cyclic voltammograms of 10 min EL copper + 1 min Pd in 0.1 M NaOH + 1 M methanol solution at different scan rates; Figure S3: Cyclic voltammograms of 10 min EL copper + 2 min Pd in 0.1 M NaOH + 1 M methanol solution at different scan rates; Figure S4: Cyclic voltammograms of 10 min EL copper + 5 min Pd in 0.1 M NaOH + 1 M methanol solution at different scan rates; Figure S5: Cyclic voltammograms of 20 min EL copper + 1 min Pd in 0.1 M NaOH + 1 M methanol solution at different scan rates; Figure S6: Cyclic voltammograms of 20 min EL copper + 2 min Pd in 0.1 M NaOH + 1 M methanol solution at different scan rates.

**Author Contributions:** Conceptualization, K.K.-S. and D.K.; methodology, K.K.-S. and A.J.; software, K.S.; validation, K.K.-S., P.Ż. and J.P.-G.; resources, K.K.-S.; data curation, K.K.-S.; writing—original draft preparation, K.K.-S.; writing—review and editing, D.K. and J.P.-G.; supervision, P.Ż. All authors have read and agreed to the published version of the manuscript.

**Funding:** This research was funded by Polish National Science Centre, grant number No UMO-2017/25/N/ST8/01721. The authors are grateful to Faculty of Non-Ferrous Metals for providing space and materials for research.

**Institutional Review Board Statement:** Not applicable.

**Informed Consent Statement:** Not applicable.

**Data Availability Statement:** Data is contained within the article or supplementary material.

**Conflicts of Interest:** The authors declare no conflict of interest. The funders had no role in the design of the study; in the collection, analyses or interpretation of data; in the writing of the manuscript, or in the decision to publish the results.

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
