# Peer review of "Well-Ordered 3D Printed Cu/Pd-Decorated Catalysts for the Methanol Electrooxidation in Alkaline Solutions"

_technologies, doi:10.3390/technologies9010006_

Round 1
Reviewer 1 Report
There are several issues in the manuscript that should be addressed before further consideration for publication.
1. Suggest the authors to include a description of the SLA or 3D printing process that is the focus of this manuscript.
- Ma et al. (2020), Metal-doped polymer-derived SiOC composites with inorganic metal salt as the metal source by digital light processing 3D printing, Virtual and Physical Prototyping 15 (3), 294-306
- Yap et al. (2020), A review of 3D printing processes and materials for soft robotics, Rapid Prototyping Journal 26 (8), 1345-1361
2. Which part of the samples are fabricated using 3D printing? The process is supposed to be layer by layer.
3. Please clarify if Figure 2 is obtained using EDS? What is the purpose of this composition analysis?
4. For the coating deposition, any analysis done on the thickness and the composition across the deposition? How will this affect the corrosion behavior?
5. Any significant changes/differences between the results from different timing? Any statistical analysis done?
6. What is the basis of the design for Figure 8? Is it based on the efficiency of the coating deposition? Any validation done?
7. Suggest to include a conclusion to summarise key findings from the studies.
Author Response
On behalf of all the co-authors, I would like to thank for the opportunity to publish the manuscript “Well-ordered 3D printed Cu/Pd-decorated catalysts for the methanol electrooxidation in alkaline solutions” in Technologies and for the Reviewers' opinions and comments. Reviews were very useful, they allowed to notice some errors in the text and to look at other aspects of issue.
The answers for reviews are presented below.
#Reviewer 1
- Suggest the authors to include a description of the SLA or 3D printing process that is the focus of this manuscript.
Ma et al. (2020), Metal-doped polymer-derived SiOC composites with inorganic metal salt as the metal source by digital light processing 3D printing, Virtual and Physical Prototyping 15 (3), 294-306
Yap et al. (2020), A review of 3D printing processes and materials for soft robotics, Rapid Prototyping Journal 26 (8), 1345-1361
Thank you for your suggestion. The bibliography has enlarged, among others on these publications.
- Which part of the samples are fabricated using 3D printing? The process is supposed to be layer by layer.
The last part of the work concerned the metallization of 3D prints produced by SLA method. In the first part, the metal coatings deposited on dropped and cured resin were analyzed to determine their properties. The next step was to deposit coatings on 3D prints.
- Please clarify if Figure 2 is obtained using EDS? What is the purpose of this composition analysis?
Yes, Figure 2 presents the EDS mapping results. The purpose of these measurements was to check if the distribution of elements is homogeneous. In the case of non-homogeneous distribution, the catalytic properties of such coatings may be difficult to define unequivocally.
- For the coating deposition, any analysis done on the thickness and the composition across the deposition? How will this affect the corrosion behavior?
Any analysis of the thickness of the coatings was not performed. In this case the most important is that the coatings are compact. Therefore, a variant of Cu metallization for 20 minutes was chosen in the case of 3D prints covering. Catalytic processes are surface processes, so this is the most important at this stage of research.
- Any significant changes/differences between the results from different timing? Any statistical analysis done?
The composition changes and thus the catalytic properties, with the time of the Pd galvanic displacement have been determined.
The R-square coefficient was determined to determine the fitting to the diffusion control model.
- What is the basis of the design for Figure 8? Is it based on the efficiency of the coating deposition? Any validation done?
The prints were designed based on common geometry, like it is presented in review: E. Bogdan, P. Michorczyk, 3D Printing in Heterogeneous Catalysis—The State of the Art, Materials (Basel). 2020 Oct; 13(20): 4534.
- Suggest to include a conclusion to summarise key findings from the studies.
Thank you for the suggestion, the text was reorganized to present the summary.
Reviewer 2 Report
The novelty of this work has been not highlighted. In my opinion the research novelty refers to the metallization of the printing elements with a large specific surface in relation to the their dimensions. The application/prospects for this elements should be discussed in introduction section and novelty aspects highlighted.
Moreover, the “discussion” section is too short. I propose to rename it into: “Conclusions” or “Summary”
Author Response
On behalf of all the co-authors, I would like to thank for the opportunity to publish the manuscript “Well-ordered 3D printed Cu/Pd-decorated catalysts for the methanol electrooxidation in alkaline solutions” in Technologies and for the Reviewers' opinions and comments. Reviews were very useful, they allowed to notice some errors in the text and to look at other aspects of issue.
The answers for reviews are presented below.
#Reviewer 2
The novelty of this work has been not highlighted. In my opinion the research novelty refers to the metallization of the printing elements with a large specific surface in relation to the their dimensions. The application/prospects for this elements should be discussed in introduction section and novelty aspects highlighted.
After careful literature overview we do not find any papers connected with direct modification of this type of photo-sensitive resin by electroless deposition and functionalization of the surface for electrochemical applications. Described method is highly-scalable and can be implemented for all desired geometries and sizes of 3D-printed elements. Electroless Cu deposition process is not complicated and further modification by Pd ions do not require high concentration of palladium metal in bath. This combination of metals provide high corrosion resistance and highly-developed surface area, what is crucial in case of any electrochemical processes.
Moreover, the “discussion” section is too short. I propose to rename it into: “Conclusions” or “Summary”
Thank you for your suggestion. The discussion is given by the individual parts of the results. The title of this section has been changed.
Reviewer 3 Report
The manuscript deals with the fabrication of 3D printed Cu/Pd-decorated catalysts for the electrooxidation of methanol to use them in direct methanol fuel cells. The manuscript presents data that make it of interest for publication, but it is not suitable for publication in the present form. Presented parts of the article as well as arrangement should be efficiently reconsidered. The manuscript needs major revision.
Comments:
- The abstract is not informative and should be rewritten.
- Line 184, the description of the catalyst prototype and experimental details of CV measurements should be put in Section 2. Materials and Methods.
- The Results section (Section 3) could be broken up into sub-sections for greater clarity like this: Physical characterization; Electrooxidation of methanol on the Cu/Pd-decorated alloys; Electrooxidation of methanol on 3D printed Cu/Pd-decorated catalysts.
- The authors must confirm that they obtained „alloys“ by XRD.
- Legends of Figures 3, 4, 5 need revision. More detailed explanations are needed. It is not clear - „10 min (a), 20 min (b) electroless deposition copper”. The prepared catalysts should be clearly presented and described. The catalysts must be named, but not the process.
- Legend of Figure 9 must be also clearly described the prepared catalysts.
- The English must be revised. Some sentences are not clear.
- The entire manuscript has some small wording and grammatical errors that can be corrected by a careful editing process.
Author Response
On behalf of all the co-authors, I would like to thank for the opportunity to publish the manuscript “Well-ordered 3D printed Cu/Pd-decorated catalysts for the methanol electrooxidation in alkaline solutions” in Technologies and for the Reviewers' opinions and comments. Reviews were very useful, they allowed to notice some errors in the text and to look at other aspects of issue.
The answers for reviews are presented below.
The manuscript deals with the fabrication of 3D printed Cu/Pd-decorated catalysts for the electrooxidation of methanol to use them in direct methanol fuel cells. The manuscript presents data that make it of interest for publication, but it is not suitable for publication in the present form. Presented parts of the article as well as arrangement should be efficiently reconsidered. The manuscript needs major revision.
Comments:
- The abstract is not informative and should be rewritten.
Thank you for your suggestion, the abstract has been improved and reorganized to make it more clear.
- Line 184, the description of the catalyst prototype and experimental details of CV measurements should be put in Section 2. Materials and Methods.
It is true that this information should be found elsewhere in the text. This has been corrected.
- The Results section (Section 3) could be broken up into sub-sections for greater clarity like this: Physical characterization; Electrooxidation of methanol on the Cu/Pd-decorated alloys; Electrooxidation of methanol on 3D printed Cu/Pd-decorated catalysts.
Thank you for the suggestion. The text was reorganized.
- The authors must confirm that they obtained „alloys“ by XRD.
The XRD analysis confirmed the coatings are amorphous. This does not allow the conclusion that the obtained coatings are alloyed. This has been corrected in the text.
- Legends of Figures 3, 4, 5 need revision. More detailed explanations are needed. It is not clear - „10 min (a), 20 min (b) electroless deposition copper”. The prepared catalysts should be clearly presented and described. The catalysts must be named, but not the process.
The descriptions for these figures have been rearranged to make them clearer. Additionally, information on the type of substrate was provided. I suppose such a description will allow a clear presentation.
- Legend of Figure 9 must be also clearly described the prepared catalysts.
Like previously, the legend was reorganized.
The English must be revised. Some sentences are not clear.
The entire manuscript has some small wording and grammatical errors that can be corrected by a careful editing process.
The text in the manuscript was checked several times for language correctness, this time it was also checked again carefully.
Round 2
Reviewer 1 Report
NA
Author Response
Thank you again for reviewing our work
Reviewer 3 Report
The authors improved their revised version of the manuscript, however, some several comments/remarks should be clarified before its final acceptance:
Line 60 “catalysts.s” change to “catalysts”
Line 73 “chromium+IV” change to “chromium(IV) or “Cr4+”
Line 77 “NaBrH4” change to “NaBH4”
Line 122: The third mapping view of “Cu”. It is seen that the presented image is an overall mapping view of Cu/Pd, not only for Cu.
Line 135, 136: “10 min (a), 20 min (b) electroless deposition copper” change to “electrolessly deposited copper for 10 (a) and 20 (b) min”
Line 138: “electroless” change to “electrolessly”
Author Response
Thank you for reviewing the manuscript, it was possible to notice errors that were corrected as follow:
Lines 60, 73, 77, 135, 136, 138 were corrected as suggested.
For mapping analysis, Figure 2 has been modified. Copper as a substrate is evenly distributed, which is certain. A scale with several colors is likely the result of too high an intensity probably. To avoid confusion or to raise doubts, the Cu distribution has been removed. In case of Cu/Pd materials, the distribution of palladium is the most important due to the homogeneity of the properties.